# Interactive Visual Reasoning under Uncertainty

**Manjie Xu** [1, *, †]
manjietsu@bit.edu.cn

**Guangyuan Jiang** [2, *]
jgy@stu.pku.edu.cn

**Wei Liang** [1, 3, ✉]
liangwei@bit.edu.cn

**Chi Zhang** [4, ✉]
zhangchi@bigai.ai

**Yixin Zhu** [2, ✉]
yixin.zhu@pku.edu.cn

* M. Xu and G. Jiang contributed equally.    ✉ corresponding authors
[1] School of Computer Science & Technology, Beijing Institute of Technology
[2] Institute for AI, Peking University
[3] Yangtze Delta Region Academy of Beijing Institute of Technology, Jiaxing, China
[4] National Key Laboratory of General Artificial Intelligence, BIGAI

**https://sites.google.com/view/ivre**

## Abstract

One of the fundamental cognitive abilities of humans is to quickly resolve uncertainty by generating hypotheses and testing them via active trials. Encountering a novel phenomenon accompanied by ambiguous cause-effect relationships, humans make hypotheses against data, conduct inferences from observation, test their theory via experimentation, and correct the proposition if inconsistency arises. These iterative processes persist until the underlying mechanism becomes clear. In this work, we devise the 🐘IVRE (pronounced as *ivory*) environment for evaluating artificial agents' reasoning ability under uncertainty. 🐘IVRE is an interactive environment featuring rich scenarios centered around *Blicket* detection. Agents in 🐘IVRE are placed into environments with various ambiguous action-effect pairs and asked to determine each object's role. They are encouraged to propose effective and efficient experiments to validate their hypotheses based on observations and actively gather new information. The game ends when all uncertainties are resolved or the maximum number of trials is consumed. By evaluating modern artificial agents in 🐘IVRE, we notice a clear failure of today's learning methods compared to humans. Such inefficacy in interactive reasoning ability under uncertainty calls for future research in building human-like intelligence.

## 1   Introduction

Situated in a room, you rarely have a clear idea of what specific factor caused a sudden lights-out. You might begin to check the light switch, the main circuit breaker, or the light bulb itself. With a series of experiments, you can finally realize the actual cause. This is a canonical example of reasoning and resolving uncertainty through interaction: when encountering a novel scenario, humans typically lack sufficient information and knowledge to arrive at a definitive conclusion based solely on the initial observation. Instead, we formulate hypotheses, subject them to testing, and utilize newly gathered data to address the preceding uncertainty in our reasoning process (Halpern, 2017).

---

†Work done while M. Xu was an intern at Peking University.

Navigating uncertainty through reasoning is a distinctive feature of human intellect. This journey, from Hume's explorations of causation (Hume, 1896) to contemporary scientific breakthroughs, illustrates humanity's reliance on formulating hypotheses to actively probe and gather fresh evidence, thereby diminishing ambiguity and carving certainties out of the realms of the unknown (White, 1990; Shanks, 1985). In the current landscape, the field of machine learning has witnessed remarkable advancements, spanning domains from comprehending natural language to deciphering visual stimuli. Yet, the aspiration to emulate human-like reasoning within machines remains an unfulfilled odyssey. The existing capabilities of artificial systems notably diverge from the human, even infantile, cognitive processes used to interpret our surroundings, deduce causal connections, and pioneer scientific discoveries (Gopnik, 1996).

In this work, we study the problem of visual reasoning under uncertainty, which tasks an agent to design new experiments that test hypotheses and discern each variable's causal role. In pursuit of this goal, we introduce the Interactive Visual Reasoning (🐘IVRE) (pronounced as *ivory*) environment as an interactive testbed. 🐘IVRE is grounded on the *Blicket* detection setup (Gopnik and Sobel, 2000; Sobel et al., 2004; Sobel and Kirkham, 2006; Zhang et al., 2021a), initially designed to evaluate children's induction ability via passive observation and active trials. In a series of experiments, children were presented with a Blicket machine, which has a very intuitive working mechanism: whenever a Blicket is put on top of it, the device becomes activated, lighting up and playing music. Participants were shown a series of experiments to demonstrate the Blicketness of a set of objects. They were then encouraged to engage in exploratory play with the objects and the machine to determine the Blicketness of each object. Critically, during the exploratory process, children generated and validated different hypotheses until they were confident about the properties of each object (Gopnik and Sobel, 2000; Sobel and

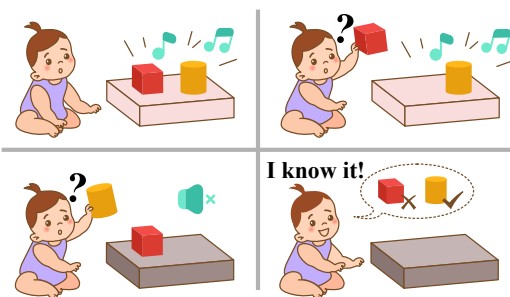

Figure 1: **Children resolve uncertainty through exploratory play with the Blicket machine and objects.** In the beginning, the child is uncertain about which object can activate the machine from the initial context panel that contains an active machine. S/he procedurally designs new experiments to test hypotheses. When a new trial indicates that the machine remains activated when only the yellow cylinder is present, the child knows that the cylinder alone can activate the machine, confirming its Blicketness. However, the red cube requires an additional test for verification.

Kirkham, 2006; Sobel et al., 2004; Walker and Gopnik, 2014) and can even rationally infer causes of failed actions when they are as young as 16-month old (Gweon and Schulz, 2011). Fig. 1 shows an illustrative example of how a participant resolves uncertainty when unraveling how the Blicket machine works and which object is a Blicket.

🐘IVRE is built following a similar interactive setting in the original Blicket experiments. Specifically, we are inspired by recent work in building synthetic *reasoning* benchmarks (Johnson et al., 2017; Edmonds et al., 2018; Zhang et al., 2019a; Yi et al., 2019; Girdhar and Ramanan, 2019; Zhang et al., 2021a; Xie et al., 2021; Li et al., 2022, 2023; Jiang et al., 2023; Xu et al., 2023) and adopt the CLEVR (Johnson et al., 2017) universe in creating the interactive environment. Following the recent work of ACRE (Zhang et al., 2021a) for causal reasoning, we borrow the object appearances and the Blicket machine setup. In particular, an agent is presented with a few observations of objects on Blicket machines that are either activated or not (referred to as *context*) at the beginning of each 🐘IVRE episode. Information for identifying which subset of objects are Blickets is incomplete from the context only, thereby introducing uncertainty. To determine the Blicketness of each object, an agent is tasked to propose trial experiments that could be carried out in each of the upcoming time steps (referred to as *trials*), and in the meantime, updating its belief over which object is an actual Blicket. The correctness of its belief at each step serves as the motivating signal for the agent, who additionally receives a constant penalty for every unsuccessful trial to encourage efficiency.

Serving as a testbed, 🐘IVRE evaluate interactive reasoning under uncertainty of today's state-of-the-art artificial agents (Schmidhuber, 2015; Lillicrap et al., 2015; Fujimoto et al., 2018; Mnih et al., 2015; Sutton and Barto, 2018; OpenAI, 2023). Not only do we benchmark Reinforcement Learning (RL) algorithms with visual input from the rendering engine, but also with the ground-truth

Table 1: Comparison between 🐘IVRE and other related visual reasoning benchmarks in terms of tasks (**cl**assification, **v**isual **q**uestion **a**nswering, and **game**), sizes (number of scenarios), and input formats. 🐘IVRE introduces a spatial-temporal-causal reasoning task, which allows intervention and belief update. It aims at few-shot uncertainty resolution with fast experimentation and reasoning.

| Benchmarks | Task | Size | Format | Temporal | Interactive | Uncertainty | Few-shot |
|---|---|---|---|---|---|---|---|
| CLEVR (Johnson et al., 2017) | vqa | 100k | image | ✗ | ✗ | ✗ | ✗ |
| CLEVRER (Yi et al., 2019) | vqa | 20k | video | ✓ | ✗ | ✗ | ✗ |
| CATER (Girdhar and Ramanan, 2019) | cls | 5.5k | video | ✓ | ✗ | ✗ | ✗ |
| CURI (Vedantam et al., 2021) | cls | 990k | image | ✓ | ✗ | ✓ | ✓ |
| ACRE (Zhang et al., 2021a) | cls | 30k | image | ✓ | ✗ | ✓ | ✓ |
| Alchemy (Wang et al., 2021) | game | - | image/symbol | ✓ | ✓ | ✓ | ✗ |
| 🐘IVRE (Ours) | game | - | image/symbol | ✓ | ✓ | ✓ | ✓ |

symbolic representation of the environment, including some additional study with Large Language Models (LLMs). We note that the environment is challenging enough for agents with even the symbolic representation, and visual complexity poses additional challenges. Further comparing the performance of the heuristic algorithms and that of the RL agents, the prominent failure of today's artificial agents in resolving uncertainty becomes even more evident, calling for future investigation into building intelligence that can learn and reason like people (Lake et al., 2017; Zhu et al., 2020).

To sum up, our work makes the following contributions:

- We present the 🐘IVRE platform, a unique environment tailored for assessing the proficiency of artificial agents in dynamically resolving uncertainty through interaction. What sets 🐘IVRE apart is the dual challenges it imposes: demanding both logical reasoning and the creation of effective strategies to mitigate uncertainty.

- The 🐘IVRE setup was meticulously designed to maintain a balance between perceptual simplicity and a rich array of visual elements and situational tasks. This environment ushers in a novel paradigm of interactive reasoning in the face of uncertainty, compelling agents to engage directly with their conjectures by formulating and executing new experimental trials.

- Utilizing the 🐘IVRE framework, we evaluated a spectrum of agents on their ability to navigate the complex problem of interactive reasoning under uncertain conditions. Additionally, our human studies confirmed the human aptitude for managing such tasks. Our observations underscore that (i). visual complexity is not the core difficulty in this task, which echos our design principle to minimize visual complexity, (ii). the art of uncertainty reduction within 🐘IVRE hinges on advanced reasoning skills and the strategic implementation of active trials, and (iii). contemporary learning agents are yet to master uncertainty reduction through interactive methods.

## 2 Related Work

**Visual Reasoning**    A range of visual reasoning and vision-language understanding tasks has been proposed recently. On the basis of a series of Visual Question Answering (VQA) benchmarks (Antol et al., 2015; Krishna et al., 2017; Tapaswi et al., 2016; Zhu et al., 2016), Johnson et al. (2017) use synthetic images depicting simple 3D shapes to scrutinize a suite of VQA models and discover their potential weaknesses. From a causal and physical reasoning perspective, the video dataset of CLEVRER was introduced by Yi et al. (2019) to investigate the performance of state-of-the-art (SOTA) models on learning complex spatial-temporal-causal structures from interacting objects in a scene. At a more abstract level, model performance in human Intelligence Quotient (IQ) tests has been studied; Barrett et al. (2018) and Zhang et al. (2019a) proposed datasets inspired by the Raven's Progressive Matrices (RPM) (Carpenter et al., 1990; Raven and Court, 1938), featuring reasoning on the hidden spatial-temporal transformation from a limited number of context panels. Approaches for this abstract reasoning task range from the neural end towards neuro-symbolism over the years (Santoro et al., 2017; Hill et al., 2018; Zhang et al., 2019b, 2021b; Wang et al., 2019; Spratley et al., 2020; Zheng et al., 2019; Wu et al., 2020). Inspired by Blicket detection and the problem of causal induction (Gopnik and Sobel, 2000; Gopnik et al., 2001), the ACRE dataset (Zhang et al., 2021a) was presented as a way to systematically evaluate current vision systems' capability in causal induction. It is worth noting a visual reasoning method based on object-centric representation and self-attention

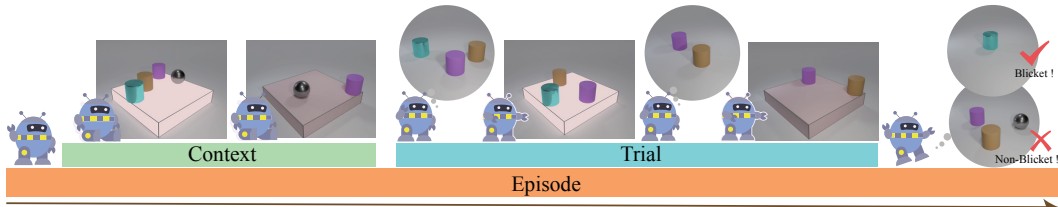

Figure 2: **A simple example episode in the 🐘IVRE environment.** At the beginning of the episode, an agent is given a set of context panels (4 in an actual instance) as an introduction to the problem. Next, the agent is motivated to determine which object is a Blicket by proposing new experiments to validate its hypothesis and update its belief. The agent will receive a high reward if all Blickets have been found out.

(Ding et al., 2021) has obtained notable performance on various visual reasoning domains, including CLEVRER (Yi et al., 2019), CATER (Girdhar and Ramanan, 2019), and ACRE (Zhang et al., 2021a), with the help of unsupervised object segmentation algorithms (Burgess et al., 2019). The inclusion of embodied versions in visual reasoning tasks also marks a significant advancement in developing AI systems that can reason within intricate and realistic environments. Some other works (Beattie et al., 2016; Wayne et al., 2018) incorporate tasks designed to evaluate the reasoning capabilities of RL agents. Hill et al. (2020) demonstrates that RL agents have the ability to conceptualize new ideas within 3D settings.

While the proposed 🐘IVRE environment is based on the Blicket setup and visually similar to the ACRE dataset, the new interactive environment emphasizes a drastically different problem: apart from figuring out the hidden causal factors, the agent is also tasked with making maximally meaningful exploration from an initial setup of high uncertainty and aggregating the collected information to update its belief and guide its next decision.

**Few-Shot Reasoning**   Several notable works have contributed to advancing agents' reasoning abilities in few-shot scenarios, where agents learn from a small number of examples or instances to solve problems. Popular works include Omniglot (Lake et al., 2015, 2019) and RAVEN (Zhang et al., 2019a). Additionally, other studies have explored the topic of uncertainty in few-shot settings. In particular, Vedantam et al. (2021) introduces a dataset that highlights compositional reasoning under uncertainty; Zhang et al. (2021a) proposes the task of causal induction, which requires a model to determine if a factor resulted in a subsequent effect; Jiang et al. (2023) introduces a benchmark to assess how machines resolve referential uncertainty when learning new words. Our environment goes beyond simple passive reasoning by allowing fast trials to reduce uncertainty.

🐘IVRE differs from previous works by introducing uncertainty into visual reasoning. Tab. 1 compares 🐘IVRE with existing dataset benchmarks. As an interactive environment, 🐘IVRE enables agents to gain information actively from the environment to test and revise their hypotheses, fundamentally distinctive from classic visual reasoning tasks such as CLEVR. The causal structure in 🐘IVRE is rich and easy for intervention, going beyond the traditional paradigm of passive observation and reasoning. In addition, 🐘IVRE also focuses on active exploration and efficiency in reasoning. Notably, while adopting a similar setting with the Blicket experiment in ACRE, 🐘IVRE is more than interactive ACRE: by abstracting out the perceptual complexity, 🐘IVRE places emphasis on the under-explored topic of uncertainty resolution. An agent needs to induce the hidden relations based on observation and, more importantly, propose interventional trials to collect new information efficiently to test its hypothesis and disentangle confounders in complex phenomena.

## 3   The 🐘IVRE Environment

We build the proposed 🐘IVRE under the OpenAI Gym framework (Brockman et al., 2016). As a visual-based interactive platform, 🐘IVRE aims at fostering an agent to understand the visual information collected and disentangle the causal factors underlying the observation by actively proposing new experiments to demystify the phenomenon, validate its hypothesis, and correct its belief. Following earlier works, the visual domain of 🐘IVRE is consistent with the CLEVR (Johnson et al., 2017) and ACRE (Zhang et al., 2021a) universe, where a Blicket machine sits on a tabletop. Lying upon the Blicket machine are objects with potential Blicketness, which can be inferred from the observation of the machine's activation pattern across frames. The objects' attributes vary in

shape (cube, sphere, or cylinder), material (metal or rubber), and color (gray, red, blue, green, brown, cyan, purple, or yellow). We signal activation of the Blicket machine by lighting it up.

An agent in 🐘IVRE is tasked with determining which objects are Blickets. At the start of each episode, the agent is presented with several initial observations of various object combinations (henceforth referred to as *context*). The context alone is insufficient to solve Blicketness for *all* objects. Hence, in each following step (henceforth referred to as *trials*), the agent proposes a new experiment of a specific object combination and updates its belief of Blicketness based on the outcome of experiments.

An episode will be terminated if the agent works out the Blicketness of *all* objects or consumes all $T = 10$ time steps. The agent is, therefore, rewarded at each step based on the correctness of its belief, and as a way to encourage efficient exploration, penalized every step it fails the problem. Please refer to Fig. 2 for a simplified example of an episode in the 🐘IVRE environment.

In the following, we discuss the designs in detail.

**Context** To instantiate a problem at each episode, we first sample 9 unique shape-material-color combinations from the pool to create the objects, the Blicketness of which is for the agent to solve. Next, we randomly assign $n$ objects ($1 \leqslant n \leqslant 4$) to be Blickets. To build a context panel, we randomly pick $m$ objects ($1 \leqslant m \leqslant 4$) out of the 9 and place them onto a Blicket machine. The status of the Blicket machine can be determined by checking if the $m$ sampled objects contain a Blicket. We repeat the sampling process 4 times to create a set of context panels as the agent's initial observation.

**Trial** After observing the context panels, the agent forms an initial belief of Blicketness over all the objects. At each time step $t$ that follows, the agent observes the outcome of the previous experiment, updates its own belief about the Blickets, and, if uncertainty about object(s) remains, proposes a new experiment to test. There is no limit to the number of objects when proposing trials. This process resembles the active hypothesis testing process and is crucial for discerning related variables.

**Observation Space** We consider two forms of observation for 🐘IVRE: a symbolic version and a pixel version. For the symbol-input version, we use a binary vector to describe the state of the scene, where the first 9 entries represent if the object of interest is present or not and the last entry if the Blicket machine is on or off. For the pixel-input version, we feed the scene description into Blender EEVEE engine (Blender Online Community, 2016) to render images in real-time. For efficiency, we render images of shape $160 \times 120$. Notably, the pixel version adds more confounding variables to the problem, whereas a well-defined symbolic version drastically simplifies it. For the pixel-input version, we hypothesize that an agent could handle uncertainty from many possible levels of abstraction; they could be defined on an attribute basis (*e.g.*, color, shape, material), an object basis, or on a group basis. We indicate the total number of Blickets for agents in both environments to improve the naive try-one-by-one strategy.

**Action Space** 🐘IVRE action space is composed of two sub-components: the trial and the belief. The trial component represents the objects selected to perform experiments on in the next trial step for resolving the uncertainty. The belief component denotes the agent's belief of Blicketness after analyzing all experimental results up to the current time step. For agents in 🐘IVRE, we soften the binary sub-spaces into continuous values in $[0, 1]$, interpreting each value in the trial sub-space as the probability of selecting that object and that in the belief sub-space as the probability of being a Blicket.

**Reward** The reward is based on the correctness of its belief and the efficiency of its trial. If the agent figures out the Blicketness for every object within the maximum $T$ time steps, it will receive a constant reward of 20. For every step that fails the guess, a penalty of $-1$ will be sent. We also introduce an auxiliary reward based on the partial correctness of its belief. Specifically, at each time step, we first calculate an oracle Blicketness belief based on all the experimental results the agent has observed via search. Next, we use the negative Jensen-Shannon Distance (JSD) between the current and oracle beliefs as the motivating signal. Note that as negative JSD is bounded by $[-1, 0]$, the reward an agent can receive ranges from $-20$ (the agent completely fails at each step and consumes all time) to 20 (the agent instantly solves the problem after observing the context).

# 4 🦣 IVRE Benchmarking

We detail two groups of models for the proposed 🦣 IVRE environment: (i). heuristic methods and (ii). RL methods under the Partially Observable Markov Decision Process (POMDP) formulation. Additionally, we collect human performance with a web-based 🦣 IVRE environment.

## 4.1 Heuristic Methods

We consider seven heuristic agents running on the symbol-input version of the 🦣 IVRE environment.

**Random Agent**    The simple random agent only randomly samples belief and the next trial from a uniform distribution without processing any observation. As a result, neither does the agent generate reasonable belief nor propose any meaningful trials during the interaction.

**Bayes Agent**    Numerous studies have shown that children and adults propose hypotheses to explain causal relationships using Bayesian models (Lucas and Griffiths, 2010; Gopnik, 1996; Gopnik and Sobel, 2000; Gopnik et al., 2001). Therefore, we propose to use a Bayes agent for evaluation as well. Specifically, we implement a Naive Bayes classifier based on the Bernoulli distribution. The classifier predicts the Blicketness for each object via the Bayes' rule using the observation collected up to this time step as training data. The agent then proposes a random trial to gather more information.

**Naive Agent**    Cook et al. (2011) reveal that, to learn a latent causal structure, children without formal science education tend to perform actions that isolate relevant variables in a naive strategy. Following this empirical observation, we implement a naive agent whose trial experiment only contains one object. In the next round, the agent updates its belief for the Blicketness of the object with certainty, and randomly selects an object that has not been tested. Though feasible and certain, this policy results in low efficiency in trials.

**NOTEARS**    NOTEARS (Zheng et al., 2018, 2020) is a score-based continuous optimization algorithm that aims at structure learning of directed acyclic graphs. It can also be used for deriving causal relations (Zhu et al., 2019). Following Zhang et al. (2021a), the causal relation learning process in 🦣 IVRE can be formulated as an optimization problem and thus learned by NOTEARS. In our NOTEARS implementation, we use a naive policy to propose trials and a nonlinear NOTEARS using MLP to calculate the belief.

**Search-based Random Agent**    We also propose a search-based random agent. The agent is non-parametric in the sense that the agent assumes knowledge of the ground-truth disjunctive causal overhypothesis and, at each time step, searches for all possible Blicket assignments that are consistent with observation up to now. The agent then computes the frequency of the appearance of an object in the set of possible Blicket assignments and randomly selects a set of objects to test.

**Search-based Naive Agent**    The search-based naive agent follows the same design as the search-based random agent in that the prior belief probability is computed in the same way. This naive version uses the probability distribution to select objects for the next round. Specifically, we apply the naive strategy and only select the object with the highest uncertainty to test.

**LLMs**    We also test contemporary LLMs on symbol-input 🦣 IVRE with GPT-3.5 (gpt-3.5-turbo) (Brown et al., 2020; Ouyang et al., 2022) and GPT-4 (gpt-4-0314) (OpenAI, 2023). As studied in previous research, LLMs demonstrate the ability to understand and reason with the context in which they operate. Specifically, we tame the LLMs using a specific template in a multi-round question-answering format. Please refer to Appx. D for additional details.

## 4.2 Reinforcement Learning Methods

The 🦣 IVRE environment can be modeled as a POMDP for RL training. Formally, the POMDP problem is a tuple $(S, A, T, R, \Omega, O, \gamma)$, where $S, A, T, R, \Omega$ are the state space, the action space, the transition probability, the reward function, and the observation space, respectively. Note that the state space covers the ground-truth belief up to each time step $t$. The action space, the observation space, and the reward function have been discussed in Sec. 3. $O$ is the observation generator for each state-action pair $(s, a)$ and $\gamma$ is the discount factor (set to 0.99). As mentioned in Sec. 3, we consider two versions of the observation space: the symbol-input version and the pixel-input version. For the

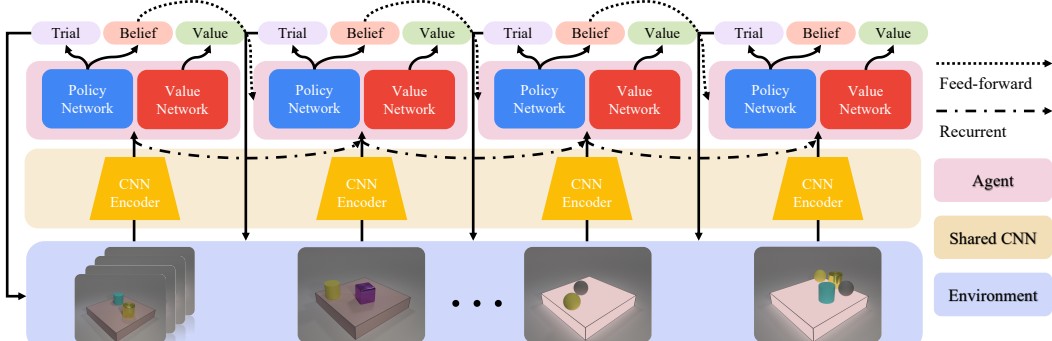

Figure 3: **The general RL architecture for benchmarking 🐘IVRE.** The architecture follows the actor-critic design; both the policy and value functions are represented with neural networks. We use a shared CNN encoder to extract visual features for the pixel-input version of the environment depicted here. The symbol-input version differs from the pixel version in providing a binary vector description processed by an MLP.

POMDP formulation of the RL algorithms, we use a recurrent agent for learning, as the underlying state is not directly observable and has to be inferred from the history of the observation.

The POMDP could also be transformed into an Markov Decision Process (MDP) if we know the underlying state at each time step. Therefore, we also consider a feed-forward architecture for this formulation: if one regards the belief sub-space from the agent's output as the true state space, we can use the belief from the previous time step and the new observation to propose a new trial and update the belief. Note that despite a feed-forward architecture being used, the entire process is still recurrent as the belief state's dependency is traced back to the history observation. See Fig. 3 for a graphical illustration of the general RL architecture we use for benchmarking 🐘IVRE. Here we only depict the pixel-input version, where the agent architecture is a CNN encoder module for visual perception and a feed-forward module that uses the agent's belief as an approximate.

As the action space has been softened, we consider popular continuous RL methods for evaluation. Specifically, we benchmark Deep Deterministic Policy Gradient (DDPG) (Lillicrap et al., 2015), Twin Delayed Deep Deterministic Policy Gradient (TD-3) (Fujimoto et al., 2018), and Proximal Policy Optimization (PPO) (Schulman et al., 2017). We also test an agent that incorporates NOTEARS (Zheng et al., 2018, 2020) as its reasoning component. Please refer to Appx. B for details.

**DDPG**  DDPG is designed based on the successful Deep Q-Learning (Mnih et al., 2015) and extended to the continuous action domain. For the symbol-input DDPG model, we use two Multi-Layer Perceptron (MLP) with three hidden layers of width $512$ and ReLU activation as the actor and critic network, respectively. For the pixel-input version, we use an ImageNet (Deng et al., 2009) pre-trained ResNet-18 (He et al., 2016) to process image panels and concatenate it with the agent's belief from the previous time step as input to the actor and critic network. We add an additional MLP and an LSTM layer with $384$ units to process the raw history observation in the recurrent agent.

**TD-3**  Compared to DDPG, TD-3 notices the problem of the overestimated value function in the actor-critic setting and suggests a new mechanism in mitigating this effect. In our TD-3 implementation, we use a backbone similar to DDPG: three MLP with three hidden layers of width $512$ and ReLU activation are used for the actor and two critic networks, respectively. A similar adaptation in the DDPG setup is made for the recurrent agent in the TD-3 implementation.

**PPO**  In RL tasks, PPO is widely used as an online learning baseline. It is based on policy optimization methods and generally has trust-region methods' stability and reliability. Similar to the two RL agents mentioned above, we adopted the same three-layer MLP for its backbone.

### 4.3  Human Baseline

We recruited 54 participants from Peking University to take part in the study, which was conducted through a web-based 🐘IVRE platform (see Fig. A8). Each participant was compensated with course credits upon the completion of an episode. The episodes were randomly allocated to the participants, who were then tasked with solving the 🐘IVRE challenge in a maximum of 10 steps, without any time restrictions.

Table 2: **Left: Performance of heuristic models on the symbol-input version of 🐘IVRE environment. Right: Performance of RL models and humans on the 🐘IVRE environment.** We report two evaluation metrics: the problem-solving accuracy, denoted as Acc, and the total reward, denoted as $R$. The context column records results when the agent only observes the context panels, whereas the episode column after the entire episode (FF: feed-forward, Re: recurrent, V: pixel-input).

| Model | Context | | Episode | | Model | Context | | Episode | |
|---|---|---|---|---|---|---|---|---|---|
| | Acc | $R$ | Acc | $R$ | | Acc | $R$ | Acc | $R$ |
| Random | 0.86% | -5.42 | 1.87% | -14.14 | DDPG-FF | 22.47% | -0.17 | 32.47% | -3.70 |
| Bayes | 15.60% | -2.98 | 43.03% | -3.78 | DDPG-Re | 13.55% | -2.03 | 46.03% | -0.29 |
| Naive | 3.50% | -3.91 | 43.62% | -1.69 | TD-3-Re | 12.57% | -2.40 | 36.83% | -2.71 |
| NOTEARS | 9.10% | -4.66 | 12.70% | -13.02 | TD-3-FF | 21.91% | -0.42 | 30.05% | -4.48 |
| Search-Naive | 1.51% | -3.68 | 83.80% | 9.39 | PPO | 6.87% | -3.87 | 28.56% | -5.85 |
| Search-Random | 1.80% | -3.62 | 34.15% | -1.87 | DDPG-V | 0.35% | -5.02 | 0.72% | -13.40 |
| GPT-3.5 | 3% | -5.91 | 11% | -13.39 | TD-3-V | 0.27% | -5.04 | 0.31% | -13.51 |
| GPT-4 | 10% | -3.36 | 26% | -7.88 | Human | 33.33% | 5.01 | 98.15% | 12.70 |

# 5 Experiment

## 5.1 Experimental Setup

We run experiments on the 🐘IVRE environment with the agents and their aforementioned variants. For evaluation metrics, we report the agent's average reward together with problem-solving accuracy over $10^4$ random test episodes (except LLMs $10^2$). The average reward measures how the model performs in terms of reasoning and trial efficiency, while the problem-solving accuracy is computed by counting the number of episodes where an agent correctly figures out Blicketness for all objects. Apart from the final results after finishing an entire episode, we also evaluate how the agent performs after observing the initial context panels only: a metric on how the model understands the problem without any trials.

## 5.2 Performance of Agents

**Heuristic Agents**   Tab. 2 shows the performance of the heuristic agents in the proposed 🐘IVRE environment. In general, we note that the more information collected, the better the agents resolve the uncertainty: the low accuracy after the context panels also verifies that additional trials are necessary for solving the entire problem. The Random agent fails in this task unsurprisingly, while the Search-Naive agent reaches more closely to the human performance. The comparison among the random agent, the naive agent, and their search-based variants indicates that both the reasoning component and the exploration strategy are beneficial for a symbolic approach. Without the reasoning component, the naive agent does not build a reliable belief over which object is more likely to be a Blicket; without the exploration strategy, an agent only randomly collects new experiments, doing little help in demystifying the phenomenon. LLM agents, equipped with priors learned from large-scale text corpora, exhibit a notable level of reasoning ability to reduce uncertainty, yet still far from human performance.

**Symbol-Input RL Agents**   Tab. 2 summarises the performance of symbol-input RL agents. RL agents performed significantly better than the Random agent, where DDPG-Re gets the highest reward. Recurrent agents generally perform better than feed-forward agents, indicating that history observation and trials play a role in generating valid hypotheses and proposing new trials. One interesting observation from the performance of these RL agents is that they generally propose very inefficient trials: the final reward decreases as time goes by. However, it becomes more likely for an agent to stumble on a solution by chance.

**Pixel-Input RL Agents**   We consider DDPG and TD-3 with the pixel-input experiments. As shown in Tab. 2, all the models tested in the pixel-input version of 🐘IVRE catastrophically fail with random-level problem-solving accuracy and total reward. Surprisingly, while their rewards are slightly higher than the random agent's, their problem-solving accuracy is even worse, let alone the conspicuously large gap from the performance under the symbol-input circumstance. We carefully checked the output of these agents and found that belief and trial predicted by agents oscillated around 0.5, indicating that they had difficulty learning to understand the 🐘IVRE environment from pixel-level

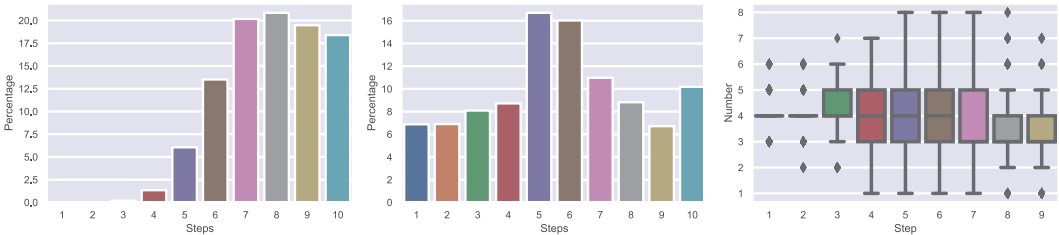

Figure 4: **Left and Middle: Distribution of the steps an agent takes to successfully solve an 🦣IVRE episode.** Left: Search-based Naive agent. Middle: DDPG-Re agent. **Right: Box plot of the number of objects the DDPG-Re agent proposes at each time step.**

input. Our results suggest that current models should be further improved in representation learning to help interactive reasoning under uncertainty.

**Comparison** Comparing experimental results in heuristic agents, symbol-input RL agents, and pixel-input RL agents, we note that uncertainty reduction is challenging not only in reasoning but also in proposing valid trials. Our heuristic results reveal that the lack of either capability can lead to failure in tasks like 🦣IVRE, which are quite common in the real world. For almost all of the RL-learned policies, they have difficulty effectively proposing meaningful experiments to validate and update their belief. The heuristic method with the best performance is designed based on human prior and equipped with a fixed and naive policy, far from the talent shown by human beings when performing this task. Even LLMs with vast prior knowledge still fall short.

## 5.3 Analysis

The naive trial policy achieves significantly better results compared to the random trial policy with oracle belief. Several factors contribute to the performance of the Search-Naive agent. First, the agent operates under the assumption that the Blicket machine functions as an OR machine (the disjointive causal overhypothesis), meaning it will activate if at least one object is a Blicket. Second, although not entirely accurate, the agent uses correlation as a proxy for causality, which may yield partially correct outcomes in certain situations. Third, the agent tests for "Blicketness" one object at a time, which, while not the most efficient approach, helps to isolate other confounding variables. These findings suggest that solving the 🦣IVRE problem requires more than random testing; it calls for thoughtful selection of trials.

Moreover, the visual complexity adds additional challenges to the problem for agents, whereas humans are used to reasoning and refining their hypotheses from visual stimuli. Looking into the failure in the pixel-input agents, we hypothesize that causal representation learning could mitigate its learning inefficacy: not only do we need features to distinguish between different experiments and those that support an in-depth understanding of the environment.

Analyzing the patterns and outcomes of human participants, we observe that they exhibit strong capabilities in reasoning and exploration. They employ a versatile approach to experimentation and demonstrate effective reasoning even within constrained scenarios. For further insights, refer to the Appx. E detailed set of examples provided.

**Distribution of Steps** The symbolic version of 🦣IVRE is considered a diagnostic tool and an upper bound for pixel 🦣IVRE. With pixel-input RL agents reaching only random-level performance, we analyze the best heuristic model (*i.e.*, Search-based Naive agent) and the best symbol-input RL model (*i.e.*, DDPG-Re). Fig. 4 shows the distribution of steps the agents take to successfully solve a problem and the number of objects proposed at each time step in the $10^4$ random test episodes. We also plot the number of objects the DDPG-Re agent proposes at each time step. The search-based method makes reasonable trials with the naive strategy at each step, and as the information collected amounts, more episodes are solved. RL agents also have learned specific exploration strategies to reduce uncertainty, although not as perfect as humans: more flexible and diverse actions are observed in Steps 5-7, and more episodes are solved in these steps. On the other hand, DDPG-Re agent might have learned data bias as it directly solves a certain number of episodes without interaction. For its trial policy, the DDPG-Re agent consistently selects around 3 to 5 objects to test at earlier steps in the process and picks fewer objects towards the end. Combining the analysis, we find the agent in

the early steps learns to use a mixed strategy even less effectively than the naive trial, showing very limited ability in actively reasoning under uncertainty.

# 6  Conclusion and Discussion

In this work, we introduce 🐘IVRE, an interactive testbed for evaluating artificial agents' reasoning ability under uncertainty. Inspired by the theoretical proposition and the empirical observation of infants, the newly introduced 🐘IVRE mimics the classic Blicket detection experiment but intentionally simplifies the sensorimotor control by abstracting it out into a discrete space of object selection. 🐘IVRE's design not only requires the agent to effectively update its belief based on the information collected so far but also necessitates the capability to come up with maximally efficient new experiments to disentangle confounding factors.

Measuring performance and concluding this manuscript is not the end but rather the beginning of our pursuit of an intelligent agent who can learn and think like people when handling uncertainty during the interaction. With today's agents catastrophically failing in this problem, how do humans, even very young children, successfully resolve uncertainty in the world around them without specific training? What role does training from nurture play in the process? And how to incubate an artificial agent with this ability through interaction with the environment? With many questions unanswered, we hope this preliminary work will motivate further research.

**Societal Implication**    We have not identified any negative societal implications arising from the proposed benchmark. On the contrary, the act of uncertainty resolution within IVRE necessitates robust reasoning capabilities, contingent on effective intervention strategies. It is noteworthy that contemporary learning agents still struggle in identifying interconnected variables through interactive engagement. We believe 🐘IVRE has the capacity to make a positive contribution towards the development of agents exhibiting human-level intelligence.

**Limitations and Future Work**    To highlight reasoning and uncertainty reduction, we design 🐘IVRE with synthetic instead of real-world scenarios. Given the challenges associated with visual perception and action in the real world, additional research is required to investigate how agents can effectively address uncertainty within a more complex environment. 🐘IVRE also employs relatively simple causal structures to create uncertainty. This calls for further research to study how machines and humans can actively resolve uncertainties from different levels in various causal structures.

**Acknowledgement**    We thank Liangru Xiang for helpful discussions, Ms. Zhen Chen (BIGAI) for designing the figures, and NVIDIA for their generous support of GPUs and hardware. M.X., G.J., W.L., C.Z., and Y.Z. are supported in part by the National Key R&D Program of China (2022ZD0114900), M.X. and W.L. are supported in part by the NSFC (62172043), and Y.Z. is in part by the Beijing Nova Program.

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

## A  Ablation with Fewer Objects

Table A1: **Performance of models on the symbolic version of 🦣IVRE with fewer objects and blickets.** We conduct experiments where at most 3 blickets are selected from 7 unique objects in 🦣IVRE. Results show that reducing the number of objects in 🦣IVRE would simplify the environment.

| Model | Random | Bayes | Naive | Search-Naive | Search-Random | DDPG-FF | TD-3-FF |
|-------|--------|-------|-------|--------------|---------------|---------|---------|
| Acc   | 7.61   | 71.92 | 94.56 | 100.00       | 54.53         | 90.08   | 88.64   |
| $R$   | -12.81 | 4.73  | 10.82 | 14.32        | 3.17          | 11.96   | 11.53   |

Striking a balance between simplicity and complexity is crucial to maintaining the 🦣IVRE's ability to assess agents' reasoning abilities effectively. We conduct experiments where at most 3 blickets are selected from 7 different objects in 🦣IVRE. The results are shown in Tab. A1. As demonstrated, reducing the number of objects in 🦣IVRE would simplify the environment. However, this approach could inadvertently lead to shortcuts, as random or naive trials might efficiently address much of the uncertainty.

## B  Agent Details

### B.1  RL Agent Details

All RL models are implemented in PyTorch (Paszke et al., 2017) under the helper library of Tianshou (Weng et al., 2022). When training the MLP and the LSTM backbones, we use the Adam optimizer (Kingma and Ba, 2014) with a learning rate of $3 \times 10^{-4}$. The DDPG and the TD-3 agent share the same hyper-parameters: we set the exploration noise to $0.1$ and the target network's soft update to $0.005$. Other parameters have been set using the default parameters from the original work. All RL models were trained for $10^7$ steps during training. The best model from training was saved and used during the evaluation phase. All the experiments reported herein were run on NVIDIA GeForce RTX 3090 or Tesla V100 graphics cards.

## C  Model Details

### C.1  Symbol-Input Backbone

Tab. A2 shows the feed-forward architecture we use for the RL agents in the symbol-input version of the environment, where the parameter of the Linear layer denotes the size of its output and $m$ is set equal to the size of the action space. Of note, the DDPG agent uses an actor network and a critic network, while the TD-3 agent uses one actor network and two critic networks. The actor network and the critic network share the same architecture. Note that the feed-forward architecture takes in both the current observation and the belief from the previous time step. Our online learning baseline PPO shares the same three-layer MLP for its backbone.

The recurrent architecture of the RL agent recruits the network components listed in Tab. A3, where we explicitly use a single-layer LSTM with a hidden size of 512 units.

Table A2: Feed-forward architecture of RL agents.

| Layer  | Units |
|--------|-------|
| Linear | 512   |
| ReLU   | /     |
| Linear | 512   |
| ReLU   | /     |
| Linear | 512   |
| ReLU   | /     |
| Linear | $m$   |

Table A3: Recurrent architecture of RL agents.

| Layer  | Units |
|--------|-------|
| LSTM   | 512   |
| Linear | 512   |
| ReLU   | /     |
| Linear | $m$   |

## C.2 Pixel-Input Backbone

The pixel-input model leverages a pre-trained ResNet-18 (He et al., 2016) for extracting image features. Specifically, we resize each image rendered from the Blender `EEVEE` engine (Blender Online Community, 2016) into $224 \times 224$ and feed it through the backbone. The features are then passed through the same actor network and the critic network listed in Tabs. A2 and A3.

## D  LLMs Experiments

We use the OpenAI Chat Completions API for our LLMs experiments; see examples in Tab. A4.

Table A4: **An example dialogue generated between GPT and our 🦣IVRE (the *User*).**

| Role | Content |
|---|---|
| User | "There are nine objects: 1, 2, 3, 4, 5, 6, 7, 8, 9 that can be put on a machine. Up to four of all the objects have a unique property that can activate the machine. Your goal is to find out all the objects that can activate the machine in ten rounds. All the following questions in the format of '1, 2, and 3.'" |
| User | "Round 0: 1 are on the machine, and the machine is not activated. Question: Which objects can activate the machine?" "Answer: From 1-9, objects that can activate the machine are" |
| Assistant | "2, 3, 4, 5, 6, " |
| User | "Your answer is wrong." |
| User | "Round 1: 3 are on the machine, and the machine is not activated. Question: Which objects can activate the machine? Answer: From 1-9, objects that can activate the machine are" |
| Assistant | "2, 4, 5, 6, 7, " |
| User | "Your answer is wrong." |
| User | "Round 2: 1, 6, 8 are on the machine, and the machine is not activated. Question: Which objects can activate the machine? Answer: From 1-9, objects that can activate the machine are " |
| Assistant | "2, 3, 4, 5, 7, " |
| User | "Your answer is wrong." |
| User | "Round 3: 3, 4, 7, 9 are on the machine, and the machine is activated. Question: Which objects can activate the machine? Answer: From 1-9, objects that can activate the machine are " |
| Assistant | "3, 4, 7, and 9." |
| User | "Your answer is wrong." |
| User | "Round 4: None are on the machine, and the machine is not activated. Question: Which objects can activate the machine? Answer: From 1-9, objects that can activate the machine are " |
| Assistant | "Please provide the objects on the machine in Round 4 for me to give..." |
| User | "Now design your own trial, you can put some of the objects on the machine to validate your hypothesis. Question: Which objects will you put on the machine? Answer: From 1-9, I will put " |
| Assistant | "2, 5, and 6 on the machine." |
| User | "Your answer is wrong." |
| User | "Round 5: 2, 5, 6 are on the machine, and the machine is not activated. Question: Which objects can activate the machine? Answer: From 1-9, objects that can activate the machine are " |

| | |
|---|---|
| Assistant | "3, 4, 7, and 9." |
| User | "Now design your own trial, you can put some of the objects on the machine to validate your hypothesis. Question: Which objects will you put on the machine? 
 Answer: From 1-9, I will put " |
| Assistant | "3 and 4 on the machine." |
| User | "Your answer is wrong." |
| User | "Round 6: 
 3, 4 are on the machine, and the machine is not activated. 
 Question: Which objects can activate the machine? 
 Answer: From 1-9, objects that can activate the machine are " |
| Assistant | "7 and 9." |
| User | "Now design your own trial, you can put some of the objects on the machine to validate your hypothesis. Question: Which objects will you put on the machine? 
 Answer: From 1-9, I will put " |
| Assistant | "7 on the machine." |
| User | "Your answer is wrong." |
| User | "Round 7: 
 7 are on the machine, and the machine is not activated. 
 Question: Which objects can activate the machine? 
 Answer: From 1-9, objects that can activate the machine are " |
| Assistant | "9." |

# E   Case Study

We give examples of interaction replay from different agents in 🐘IVRE in Figs. A1 to A7. In each case, the first two figures show available objects in an episode, where the first one contains Blickets and the second one non-Blickets. For the naive agent, it always proposes a trial composed of a single object (see Fig. A1), and from its result of activation, the agent deterministically assigns Blicketness to an object. A search-based naive agent proposes trials based on the confidence of every object. Figs. A3 and A4 show trials from such an agent. In Fig. A3, the agent is quite sure of the non-Blicketness of the purple and green metal cube after the context panels, so it tests other objects. However, it cannot figure out the Blicketness of the green metal ball through all of the contexts, leading to its failure.

The DDPG agent, however, has unique behavior. In Fig. A6, all the Blicket machines in initial contexts are activated, so the agent has no chance to generate correct belief from the context only, except guessing. However, the DDPG agent makes repetitive trials, making it harder to gain effective information to solve the problem. In follow-up experiments, the DDPG agent also solves tasks when most objects are non-Blickets. But it often fails when there are more Blickets, which means it cannot handle situations with high correlation.

Humans show powerful abilities in both reasoning and exploration. Fig. A7 shows an example where a successful trial is achieved. We can see that a flexible trial policy is adopted, and humans can reason effectively given limited contexts.

# F   Data Documentation

We follow the datasheet proposed in Gebru et al. (2021) for documenting our proposed benchmark:

1. **Motivation**

   (a) **For what purpose was the benchmark created?**
       The benchmark was created as an environment for evaluating artificial agents' reasoning ability under uncertainty. 🐘IVRE is an interactive environment featuring rich scenarios centered around Blicket detection. Agents in 🐘IVRE are placed into environments with various ambiguous action-effect pairs and asked to figure out each object's role. Agents are encouraged to propose effective and efficient experiments to validate their

hypotheses based on observations and gather more information. The game ends when all uncertainties are resolved or the maximum number of trials is consumed.

(b) **Who created the dataset and on behalf of which entity?**
This dataset was created by Manjie Xu, Guangyuan Jiang, Wei Liang, Chi Zhang and Yixin Zhu. They are from Beijing Institute of Technology (Manjie Xu, Wei Liang), Peking University (Guangyuan Jiang, Yixin Zhu) and National Key Laboratory of General Artificial Intelligence, BIGAI (Chi Zhang).

(c) **Who funded the creation of the dataset?**
M.X., G.J., W.L., C.Z., and Y.Z. are supported in part by the National Key R&D Program of China (2022ZD0114900), M.X. and W.L. are supported in part by the NSFC (62172043), and Y.Z. is in part by the Beijing Nova Program.

(d) **Any other Comments?**
None.

2. **Composition**

(a) **What do the instances that comprise the benchmark represent?**
The benchmark contains episodes in which agents are tasked to figure out which objects are Blickets.

(b) **How many instances are there in total?**
N/A. Each episode in 🐘IVRE is randomly sampled and can have infinite tasks.

(c) **Does the dataset contain all possible instances or is it a sample (not necessarily random) of instances from a larger set?**
🐘IVRE contains all possible instances.

(d) **What data does each instance consist of?**
In each episode, an agent is presented with several initial observations of various object combinations (referred to as *context*). The context alone is insufficient to solve Blicketness for *all* objects. Hence, in each following step (referred to as *trials*), the agent proposes a new experiment of a specific object combination and updates its belief of Blicketness based on the outcome of experiments.

(e) **Is there a label or target associated with each instance?**
Yes.

(f) **Is any information missing from individual instances?**
No.

(g) **Are relationships between individual instances made explicit?**
Yes.

(h) **Are there recommended data splits?**
No.

(i) **Are there any errors, sources of noise, or redundancies in the benchmark?**
No.

(j) **Is the benchmark self-contained, or does it link to or otherwise rely on external resources (*e.g*., websites, tweets, other datasets)?**
Self-contained.

(k) **Does the benchmark contain data that might be considered confidential (*e.g*., data that is protected by legal privilege or by doctor-patient confidentiality, data that includes the content of individuals' non-public communications)?**
No.

(l) **Does the benchmark contain data that, if viewed directly, might be offensive, insulting, threatening, or might otherwise cause anxiety?**
No.

(m) **Does the benchmark relate to people?**
No.

(n) **Does the benchmark identify any subpopulations (*e.g*., by age, gender)?**
No.

(o) **Is it possible to identify individuals (*i.e*., one or more natural persons), either directly or indirectly (*i.e*., in combination with other data) from the dataset?**
No.

(p) **Does the dataset contain data that might be considered sensitive in any way (*e.g.*, data that reveals racial or ethnic origins, sexual orientations, religious beliefs, political opinions or union memberships, or locations; financial or health data; biometric or genetic data; forms of government identification, such as social security numbers; criminal history)?**
No.

(q) **Any other comments?**
None.

3. **Collection Process**

(a) **How was the data associated with each instance acquired?**
We render them using Blender.

(b) **What mechanisms or procedures were used to collect the data (*e.g.*, hardware apparatus or sensor, manual human curation, software program, software API)?**
In each episode, 🦣IVRE will randomly sample objects and blickets from a pool.

(c) **If the dataset is a sample from a larger set, what was the sampling strategy (*e.g.*, deterministic, probabilistic with specific sampling probabilities)?**
N/A.

(d) **Who was involved in the data collection process (*e.g.*, students, crowdworkers, contractors) and how were they compensated (*e.g.*, how much were crowdworkers paid)?**
Manjie Xu and Guangyuan Jiang wrote the generation code.

(e) **Over what timeframe was the data collected?**
N/A.

(f) **Were any ethical review processes conducted (*e.g.*, by an institutional review board)?**
The dataset raises no ethical concerns.

(g) **Does the dataset relate to people?**
No.

(h) **Did you collect the data from the individuals in question directly, or obtain it via third parties or other sources (*e.g.*, websites)?**
N/A.

(i) **Were the individuals in question notified about the data collection?**
N/A.

(j) **Did the individuals in question consent to the collection and use of their data?**
N/A.

(k) **If consent was obtained, were the consenting individuals provided with a mechanism to revoke their consent in the future or for certain uses?**
N/A.

(l) **Has an analysis of the potential impact of the dataset and its use on data subjects (*e.g.*, a data protection impact analysis) been conducted?**
Yes.

(m) **Any other comments?**
None.

4. **Preprocessing, Cleaning and Labeling**

(a) **Was any preprocessing/cleaning/labeling of the data done (*e.g.*, discretization or bucketing, tokenization, part-of-speech tagging, SIFT feature extraction, removal of instances, processing of missing values)?**
N/A.

(b) **Was the "raw" data saved in addition to the preprocessed/cleaned/labeled data (*e.g.*, to support unanticipated future uses)?**
N/A.

(c) **Is the software used to preprocess/clean/label the instances available?**
N/A.

(d) **Any other comments?**
None.

5. **Uses**

   (a) **Has the dataset been used for any tasks already?**
   No, the dataset is newly proposed by us.

   (b) **Is there a repository that links to any or all papers or systems that use the dataset?**
   Yes, we provide the link to all related information on our project website.

   (c) **What (other) tasks could the dataset be used for?**
   This dataset could be used for other reserach topics like causal discovery, causal reasoning and active learning.

   (d) **Is there anything about the composition of the dataset or the way it was collected and preprocessed/cleaned/labeled that might impact future uses?**
   N/A.

   (e) **Are there tasks for which the dataset should not be used?**
   N/A.

   (f) **Any other comments?**
   None.

6. **Distribution**

   (a) **Will the dataset be distributed to third parties outside of the entity (*e.g.*, company, institution, organization) on behalf of which the dataset was created?**
   No.

   (b) **How will the dataset be distributed (*e.g.*, tarball on website, API, GitHub)?**
   🐘IVRE could be accessed on our project website.

   (c) **When will the dataset be distributed?**
   🐘IVRE has already been released.

   (d) **Will the dataset be distributed under a copyright or other intellectual property (IP) license, and/or under applicable terms of use (ToU)?**
   We release our benchmark under CC BY-NC [1] license.

   (e) **Have any third parties imposed IP-based or other restrictions on the data associated with the instances?**
   No.

   (f) **Do any export controls or other regulatory restrictions apply to the dataset or to individual instances?**
   No.

   (g) **Any other comments?**
   None.

7. **Maintenance**

   (a) **Who is supporting/hosting/maintaining the dataset?**
   Manjie Xu and Guangyuan Jiang are maintaining.

   (b) **How can the owner/curator/manager of the dataset be contacted (*e.g.*, email address)?**
   manjietsu@bit.edu.cn, jgy@stu.pku.edu.cn

   (c) **Is there an erratum?**
   Future erratum will be released through the website.

   (d) **Will the dataset be updated (*e.g.*, to correct labeling errors, add new instances, delete instances')?**
   Yes.

   (e) **If the dataset relates to people, are there applicable limits on the retention of the data associated with the instances (*e.g.*, were individuals in question told that their data would be retained for a fixed period of time and then deleted)?**
   N/A. The dataset does not relate to people.

   (f) **Will older versions of the dataset continue to be supported/hosted/maintained?**
   Yes.

---

[1] https://creativecommons.org/licenses/by-nc/4.0/

(g) **If others want to extend/augment/build on/contribute to the dataset, is there a mechanism for them to do so?**
Yes. We have released the source code as well as a licence on our project website. Future developments are welcome.

(h) **Any other comments?**
None.

Blicket    Non-Blicket    Initial Contexts:

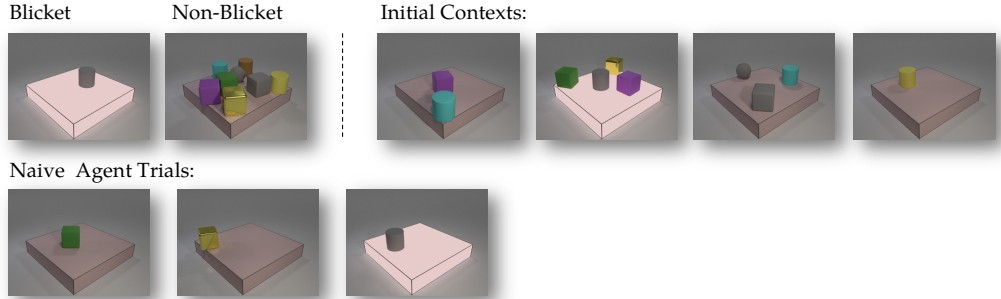

Naive  Agent Trials:

Figure A1: An example that is solved by the naive agent.

Blicket    Non-Blicket    Initial Contexts:

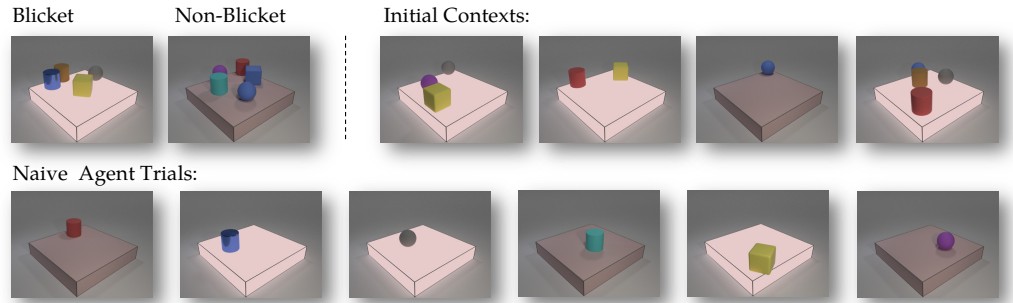

Naive  Agent Trials:

Figure A2: An example where the naive agent fails.

Blicket    Non-Blicket    Initial Contexts:

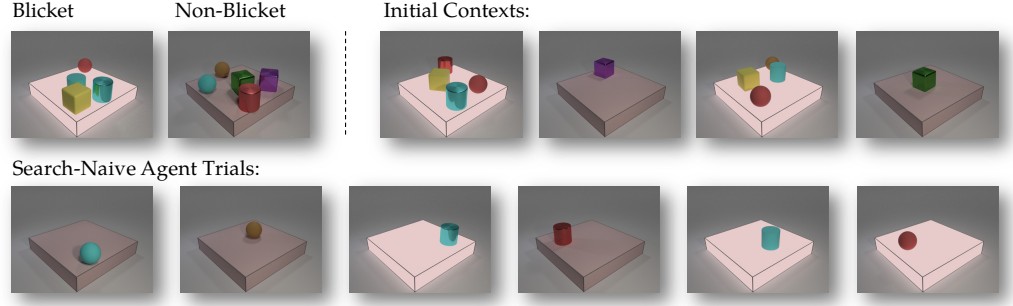

Search-Naive Agent Trials:

Figure A3: An example that is solved by the search-naive agent.

Blicket    Non-Blicket    Initial Contexts:

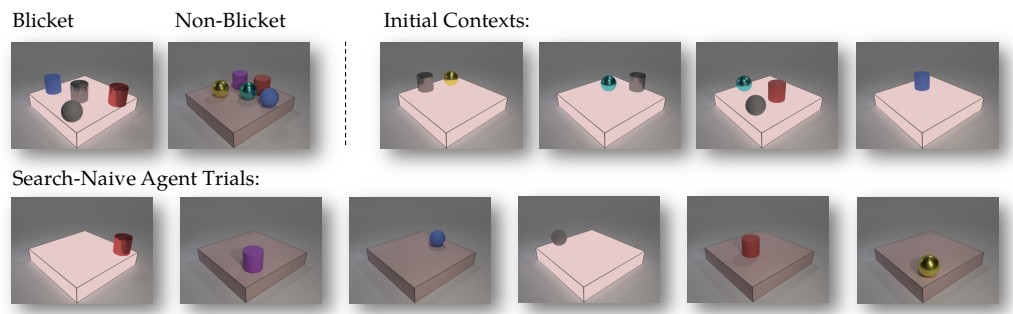

Search-Naive Agent Trials:

Figure A4: An example where the search-naive agent fails.

Blicket    Non-Blicket    Initial Contexts:

DDPG Agent Trials:

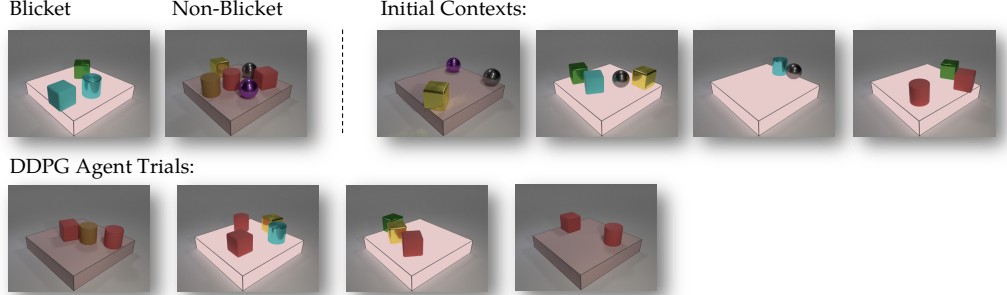

Figure A5: An example that is solved by the DDPG agent.

Blicket    Non-Blicket    Initial Contexts:

DDPG Agent Trials:

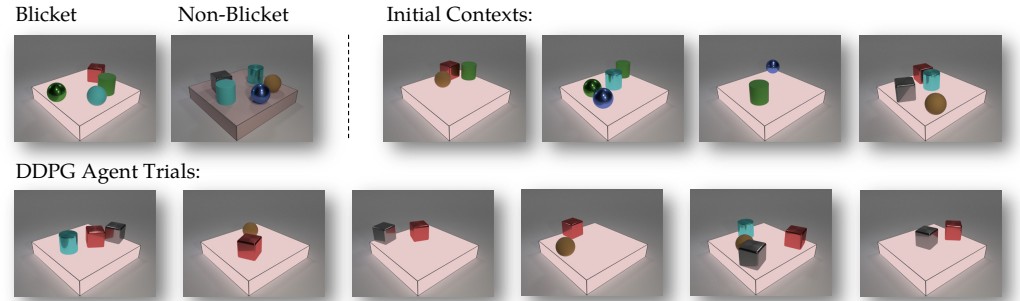

Figure A6: An example where the DDPG agent fails.

Blicket    Non-Blicket    Initial Contexts:

Human Trials:

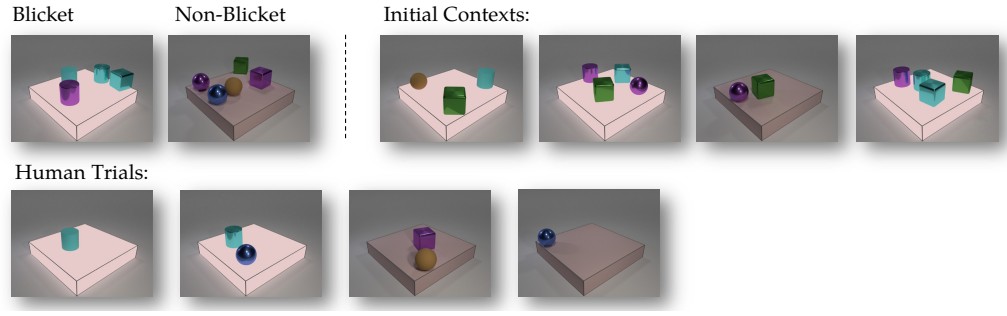

Figure A7: An example that is solved by a human participant.

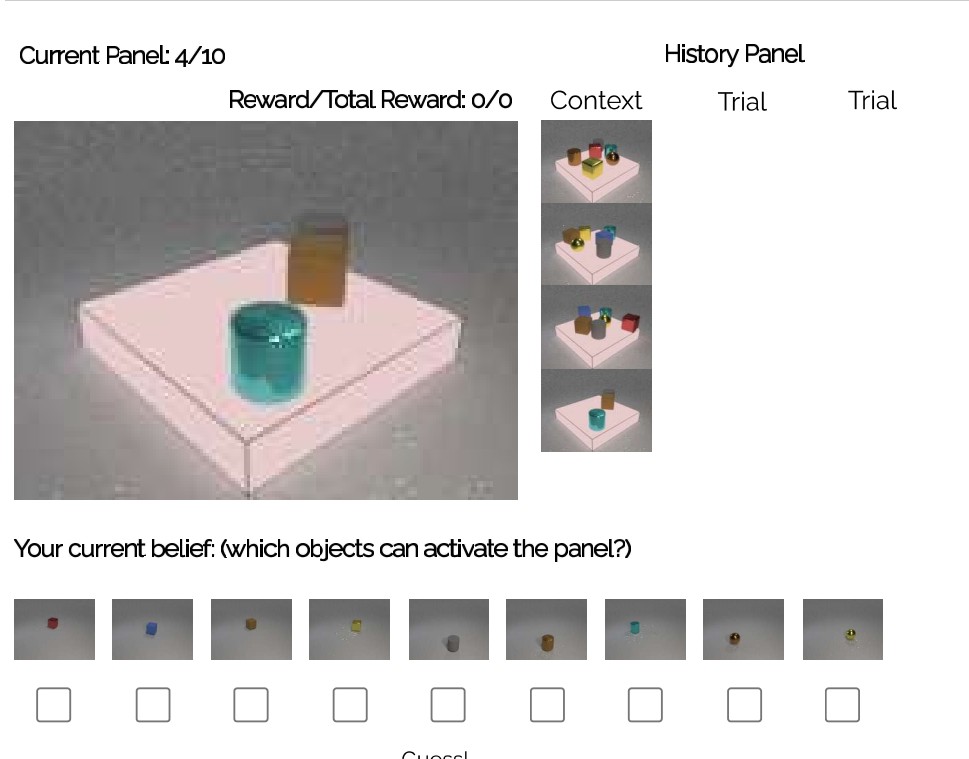

Figure A8: UI design for the web-based 🐘IVRE environment.

