# OpenReview forum: "Interactive Visual Reasoning under Uncertainty"
_NeurIPS.cc/2023/Track/Datasets_and_Benchmarks — NeurIPS 2023 Datasets and Benchmarks Poster_

### Official Review · Reviewer_pTW5 · 2023-07-17

**Rating:** 7
**Confidence:** 4
**Clarity:** Good.

**Strengths:**

- The paper is well written.
- Code and document are provided on the website.
- The idea of incorporating visual reasoning into a sequential decision-making process is novel.
- Evaluation is comprehensive and informative.

**Additional Feedback:**

See above.

**Correctness:**

Are results from table 2 from multiple independent runs? Is the performance difference statistically significant?

**Documentation:**

Fair. Authors are encouraged to host docs on a dedicated page.

**Limitations:**

- L98 authors mention the simplicity but not lack of sensorimotor control. However, the action space of the environment is merely probability simplex over objects, which is totally different from the common definition of motor control in reinforcement learning or robotics. Therefore, it is not faithful to to call it "sensorimotor control" at all.

- More technical aspects about the proposed environment are encouraged to include. For example, since running RL requires significant data collection, what is the throughput for both symbol-based and pixel-baed inputs?

**Opportunities For Improvement:**

It is interesting to further scale up the complexity of the benchmark.

**Relation To Prior Work:**

It is worth discussing embodied versions of visual reasoning tasks. For example, DMLab-30 [Beattie et al., 2016] includes tasks to test RL agents' reasoning ability [Wayne et al., 2018]. Hill et al., 2020 shows that RL agents can reason about novel concepts in 3D environments. VIMABench [Jiang et al., 2022] develops multiple visual reasoning tasks for robotic manipulation and learns a single agent to solve them.

References
-  Beattie et al., DeepMind Lab, 2016.
- Wayne et al., Unsupervised Predictive Memory in a Goal-Directed Agent, 2018.
- Hill et al., Grounded Language Learning Fast and Slow, 2020.
- Jiang et al., VIMA: General Robot Manipulation with Multimodal Prompts, 2022.

**Summary And Contributions:**

The paper proposes IVRE, an interactive visual reasoning environment to evaluate agent's reasoning ability under uncertainty. It is inspired by the Blicket experiment and adopts CLEVR in environment creation. Comprehensive evaluation is performed to serve as baseline results on this environment.

---

> ### Author Response · Authors · 2023-08-21
> **Rebuttal to reviewer pTW5**
>
> Dear reviewer pTW5,
>
> Thank you very much for your insightful and detailed review!
>
> > It is interesting to further scale up the complexity of the benchmark.
>
> Thank you for providing us with your suggestion, and we genuinely appreciate your enthusiasm for our future endeavors. We are indeed exploring intriguing avenues for future work. Among these directions, we are considering the incorporation of diverse hierarchical causal structures as well as the introduction of additional confounding factors like color and shape. We would also consider distributed solutions for reinforcement learning to further scale up.
>
> > L98 authors mention the simplicity but not lack of sensorimotor control. However, the action space of the environment is merely probability simplex over objects, which is totally different from the common definition of motor control in reinforcement learning or robotics. Therefore, it is not faithful to to call it "sensorimotor control" at all.
>
> Thank you for pointing it out. This is a mistake in the manuscript and we will correct this statement in the revised version.
>
> > More technical aspects about the proposed environment are encouraged to include. For example, since running RL requires significant
> data collection, what is the throughput for both symbol-based and pixel-baed inputs?
>
> Thank you for your suggestion. Rendering plays a substantial role in determining the throughput, with most of it being bottlenecked by this process. We conducted IVRE tests on an RTX 3090 GPU workstation paired with an AMD Ryzen 9 5950X CPU. The symbol-based IVRE achieves approximately 6000 panels per second, while the pixel-based IVRE lags behind at 50 panels per second. This disparity arises from the on-the-fly rendering required for each figure in pixel-based IVRE, leading to a slower processing rate. These specific hardware and throughput details will be included in the supplementary section of the revised paper for transparency.
>
> > Independent runs
>
> Thanks for the question. Table 2 is from multiple independent runs. As the number of runs increases, the difference will be more and more statistically significant.
>
> > Relation To Prior Work:
>
> Thank you for your insightful suggestion. We will certainly incorporate a paragraph in the related work section to discuss the relationship between IVRE and the mentioned works. The inclusion of embodied versions in visual reasoning tasks marks a significant advancement in developing AI systems that can reason within intricate and realistic environments. The cited works demonstrate the intricate interplay between perception, action, and reasoning. Nonetheless, our proposed IVRE benchmark differentiates itself from these approaches in that it necessitates active experimentation and trials to address uncertainties. IVRE not only presents diverse causal structures but also prioritizes simplicity, providing a unique perspective on reasoning challenges. This distinction solidifies IVRE's contribution to the field of active reasoning under uncertainty.

---

> > ### Comment · Reviewer_pTW5 · 2023-08-21
> >
> > Thanks authors for addressing my questions. The submission will become stronger after including the above discussion into the camera-ready version. I'd also suggest authors to host code and docs on dedicated pages instead of through Google drive, which can make the work more accessible to the community. Overall, I'm happy to raise my score from 6 to 7.

---

> > > ### Author Response · Authors · 2023-08-22
> > >
> > > Thank you for your kind suggestion! We will host all docs, code and checkpoints on a seperate github page together with the final version.

---

### Official Review · Reviewer_7bBX · 2023-07-18
**A benchmark for cognitive reasoning.**

**Rating:** 7
**Confidence:** 4

**Strengths:**

This paper proposes a new task that emphasises an interesting problem for the community (research in the field of reasoning, including visual reasoning). The interesting factor is introducing uncertainty to the reasoning process forcing an agent to formulate and test hypotheses in the given scenario. I believe this to be an interesting problem for the community being relevant to various exploratory scenarios.

This work builds the environment upon CLEVR [1] in terms of visual representation, making it easily reproducible.

A rich benchmark of various models is introduced. I believe that a good range and selection of models are tested. I appreciate benchmarking on basic, heuristic methods (random, naive, etc.) as a baseline, as well as a comparison to current LLMs and RL models, followed by a human benchmark. The results provided prove that compositional reasoning introduced as in this benchmark is indeed a challenging task for most of the current models (yet not completely unsolvable).

**Additional Feedback:**

References:

[1] Johnson, J., Hariharan, B., Van Der Maaten, L., Fei-Fei, L., Lawrence Zitnick, C., and Girshick, R. (2017). Clevr: A diagnostic dataset for compositional language and elementary visual reasoning. In Conference on Computer Vision and Pattern Recognition (CVPR).

[2] Yi, K. and Wu, J. and Gan, C. and Torralba, A. and Kohli, P. and Tenenbaum, J. B. (2018). Neural-symbolic VQA: Disentangling reasoning from vision and language understanding. In Advances in Neural Information Processing Systems.

[3] Locatello, F. and Weissenborn, D. and Unterthiner, T. and Mahendran, A. and Heigold, G. and Uszkoreit, J. and Dosovitskiy, A. and Kipf, T. (2020). Object-Centric Learning with Slot Attention. In Advances in Neural Information Processing Systems.

**Clarity:**

The paper is clearly written, easy to follow.


**Correctness:**

The benchmark is designed in a sound way and offers a good selection of models in experimental section.


**Documentation:**

This is a benchmark paper and details on creating episodes are provided in the paper and attached benchmark card. It is implemented in a very popular toolkit within the research community, thus not raising concerns about usability. I expect authors to release checkpoints for all benchmarked methods if the paper gets accepted.

**Ethics:**

No ethical concerns.


**Limitations:**

I believe authors correctly identified the main limitation and future work - i.e. introducing more visually complex environment, and further real-world scenarios. As mentioned above visual part of pixel-input models seems to be critical and more investigation into that would be great to see.

Societal impact is not discussed, which I belive is a requirement of the conference. It would be good if authors could add that.

**Opportunities For Improvement:**

From the benchmarking table (Tab.2) we can see a drastic drop in performance w.r.t. symbol-input agents. It would be good to see more investigation into the performance of such agents. Specifically, NS-VQA [2] shows that regular visual reasoning on CLEVR is nearly a solved task, hence, such a model could be used in comparison. In fact, NS-VQA translates an image into a fully disentangled representation, hence would be compatible with symbol-input agents. This could be a good bridge between symbol-input and pixel-input models in the table. Similarly, one could test a slot-attention [3] based model which extracts object-level features, on the task with a similar intention of assessing the impact of visual recognition of the agent.

The images in the benchmark are of rather small size and with many objects. It would be useful to see an ablation with a bigger image size to show how that impacts pixel-input models.

The paper mentions that context only does not provide complete information about *blicketness*. How is that assured?

It would also be good to report agents' performance with a fixed seed.

Another opportunity for improvement would be introducing an ablation with a lower number of objects in the scene.

With respect to the generation of the episodes, it would be useful to see what is the average number of trials needed to solve an episode (by an oracle) - it would provide more meaning to reported reward values. Similarly, it would be interesting to see the average reward for successful episodes for different agents, i.e. compare if they reach the solution equally fast.

**Relation To Prior Work:**

I believe this work appropriately describes prior work and places itself in its context.


**Summary And Contributions:**

This work formulates a task and benchmark for Interactive Visual Reasoning. It involves proposing a new interactive environment for evaluating agents. The goal of the task is to find all *blickets* in the scene, where *blicket* is an object triggering a response from the environment when being active. The goal of the task is to perform reasoning under uncertainty through giving the agent the possibility of testing hypotheses in the environment. Furhter, this work performs benchmarking of various agents to emphasise challanges of the proposed task.

---

> ### Author Response · Authors · 2023-08-21
> **Rebuttal to reviewer 7bBX (part 1)**
>
> Dear reviewer 7bBX,
>
> Thank you for your insightful review!
>
> > From the benchmarking table (Tab.2) we can see a drastic drop in performance w.r.t. symbol-input agents. It would be good to see more investigation into the performance of such agents. Specifically, NS-VQA [2] shows that regular visual reasoning on CLEVR is nearly a solved task, hence, such a model could be used in comparison. In fact, NS-VQA translates an image into a fully disentangled representation, hence would be compatible with symbol-input agents. This could be a good bridge between symbol-input and pixel-input models in the table. Similarly, one could test a slot-attention [3] based model which extracts object-level features, on the task with a similar intention of assessing the impact of visual recognition of the agent.
>
> Thank you for your insightful suggestion. As our work focuses on ***interective*** reasoning, ie, proposing new trials to gather information and based on the new information making further decisions, the NS-VQA framework for langauge based static QA is not really compatible. We have tried a slot-attention-based method, using GT object masks and replacing the visual feature extractor with Slot Attention Transformer model. The model yielded limited improvements (reward -13.89%, accuracy 0.62%) compared to other visual models. Note that the symbolic baselines could be considered as an upper bound for the object-centric approach, as each object is represented as a one-hot vector.
>
> > The images in the benchmark are of rather small size and with many objects. It would be useful to see an ablation with a bigger image size to show how that impacts pixel-input models.
>
> Thank you very much for your valuable suggestion. Currently, the image size we use in pixel-input models is 160*120, which is close to a common practice (e.g., both imagenet & resnet inputs are 224*224). We have tried 640*480 input but there is no significant performance gap.
>
> > The paper mentions that context only does not provide complete information about blicketness. How is that assured?
>
> Note that there are at most 4 blickets among all 9 objects. Comprehensive information is solely attainable when ***all four blickets are individually featured in the four context panels and they all activate the machines***. This case is just rejected during sampling.
>
> > It would also be good to report agents' performance with a fixed seed.
>
> For all RL models, we test all of these models with seed 0.
>
> > Another opportunity for improvement would be introducing an ablation with a lower number of objects in the scene.
>
> Thanks for the suggestion. We show IVRE results where at most 3 blickets are selected from 7 different objects below:
> | Model | Random  | Bayes | Naive | Search-Naive | Search-Random | DDPG-FF | TD-3-FF |
> | - | -| - | - | -- | - | - | ------- |
> | Acc   | 7.61    | 71.92 | 94.56 | 100.00       | 54.53         | 90.08   | 88.64   |
> | R     | \-12.81 | 4.73  | 10.82 | 14.32        | 3.17          | 11.96   | 11.53   |
>
> As the results demonstrate, reducing the number of objects in IVRE would simplify the environment. However, this approach could inadvertently lead to shortcuts, as random or naive trials might efficiently address much of the uncertainty. Striking a balance between simplicity and complexity is crucial to maintain the benchmark's ability to assess agents' reasoning abilities effectively.
>
> > With respect to the generation of the episodes, it would be useful to see what is the average number of trials needed to solve an episode (by an oracle) - it would provide more meaning to reported reward values.
>
> From our perspective, identifying an oracle solution for the proposed IVRE benchmark proves to be a challenging task; we can hardly see a ***general*** oracle solution for the proposed IVRE (see response to hE8F). To recap:
> IVRE needs agents to propose efficient trials to resolve uncertainty in the current episode. The hypothesis space can be seen as a tree with $ 2^{objnum} $ branches. And aprior you know there are $ n $ blickets in total. In each step, we can prune some branches based on the new information and the known number of blickets. The most related math problem is Balance Puzzle. However, our problem differs from the puzzle in that there could be an arbitrary number of blicketks rather than 1, and that we do not need an equal number of objects to test in each step. By manual inspection, we also note that the full search process would be long and short for configurations with only minor differences, given different context. We believe the search process is ad-hoc. In human experiments, participants tend to employ an approximative version of extensive search, strongly relying on heuristics.
> In the case of our most proficient model, Search-Naive, it takes an average of 5.51 steps to resolve an IVRE episode, yielding an average reward of 13.14. These metrics collectively reflect the intricacy and dynamic nature of the IVRE environment.

---

> ### Author Response · Authors · 2023-08-21
> **Rebuttal to reviewer 7bBX (part 2)**
>
> > Similarly, it would be interesting to see the average reward for successful episodes for different agents, i.e. compare if they reach the solution equally fast.
>
> We respectfully disagree that the successful rewards are more meaningful, compared to the total rewards. We have noted in our experiments that agents showing low total rewards tend to "solve" a problem pretty fast, compared to higher-total-reward agents. It's more likely they overfit to the most likely guess rather than mastering the process, creating the illusion that agents achieving low rewards are fast learners. The total rewards better reflect the nuanced dynamics between different agents' strategies and their outcomes.
>
> > Societal impact
>
> We are deeply grateful for your suggestion, and we are committed to incorporating a section on Societal Impact in our revised version. After a thorough review of the proposed IVRE and its potential applications, we have not identified any negative societal implications arising from the dataset. On the contrary, the act of uncertainty resolution within IVRE necessitates robust reasoning capabilities, contingent on effective intervention strategies. It's noteworthy that contemporary learning agents still struggle in identifying interconnected variables through interactive engagement. We firmly believe that this benchmark has the capacity to make a positive contribution towards the development of agents exhibiting human-level intelligence.

---

> > ### Comment · Reviewer_7bBX · 2023-08-23
> > **Reviewer response**
> >
> > Thank you for your response. I believe all my concerns were addressed by the authors.
> >
> > I appreciate adding the slot-attention experiment. It is an interesting addition in my opinion and I agree with the authors that it can be compared to symbolic baselines treating them as upper bound, therefore giving us some insight of the importance of the visual recognition part of the task.
> >
> > Thanks for the ablation with the higher resolution. Whereas I understand that your choice of resolution was probably motivated in part with rendering speed, I think this ablation is useful in proving that such a resolution is a viable choice and does not skew the results for pixel-input models.
> >
> > Thank you as well for 3 blickets experiment, it shows that the complexity of your benchmark was chosen appropriately.
> >
> > All other questions were also answered in detail. Therefore, I currently have no further concerns.

---

> > > ### Author Response · Authors · 2023-08-24
> > >
> > > Thank you for taking the time to review our manuscript and for providing your insightful feedback. We are pleased to hear that your concerns have been satisfactorily addressed. Your engagement and thoughtful insights have been instrumental in enhancing the quality of our work.

---

### Official Review · Reviewer_KESZ · 2023-07-21
**Useful contribution to ambiguity resolution, some aspects of paper unclear**

**Rating:** 7
**Confidence:** 3
**Correctness:** Yes.

**Strengths:**

-	The authors intelligently adapt the CLEVR environment to produce a benchmark for vision-centric ambiguity resolution.
-	The paper includes a strong experiments section that considers a wide range of approaches. It is helpful that human performance is included in this section. The inclusion of GPT is interesting.
-	The metrics used for the evaluation are well-chosen and informative.
-	Two versions of the task are presented: A symbolic version and a pixel version. This flexibility allows for the visual reasoning portion of the task to be abstracted to observe reasoning ability alone, which is helpful given the complexity of the overall task.

**Additional Feedback:**

N/A

**Clarity:**

The paper is generally well-written. A more explicit description of the task itself and the nature of the dataset would be helpful, especially in the introduction and Section 3.

**Documentation:**

Yes. The code is well-presented and it is nice that they include materials to host a flask app for the system.

**Limitations:**

Yes. The limitations section is comprehensive.

**Opportunities For Improvement:**

-	There are a few issues of clarity in the paper. To new audiences, the task is not detailed in a way that is easy to interpret without prior knowledge of the ACRE paper. What exactly does it mean to “propose an experiment”? This is answered later in the text, but it is confusing when mentioned in the introduction.
-	It would be nice to include a discussion of the performance gap between search-naïve and the human performance.
-	The environment is not terribly novel, as it largely adopts the framework of ACRE.
-	Some more analysis regarding the very low visual environment performance would be helpful. While lower performance is to be expected, the numbers reported in the experiments section are surprisingly low.

**Relation To Prior Work:**

The paper appears to include an appropriate list of citations.

**Summary And Contributions:**

The paper introduces a novel visual reasoning environment for assessing the interactive uncertainty resolution abilities of AI systems using a Blicket-inspired task requiring an agent to form intelligent hypotheses based on observations and efficiently test these hypotheses. The environment can be deployed as a symbolic environment or as a pixel-wise, visual environment. They evaluate a selection of approaches on this environment, both in the symbolic and visual domains, and compare these approaches to human performance on the task.

---

> ### Author Response · Authors · 2023-08-21
> **Rebuttal to reviewer KESZ**
>
> Dear reviewer KESZ,
>
> Thank you very much for your detailed and thoughtful review!
>
> > There are a few issues of clarity in the paper. To new audiences, the task is not detailed in a way that is easy to interpret without prior knowledge of the ACRE paper. What exactly does it mean to “propose an experiment”? This is answered later in the text, but it is confusing when mentioned in the introduction.
>
> Thank you for bringing this to our attention! We fully acknowledge the need for a more intuitive introduction to the blicket setting, and we're committed to addressing this in the revised version. Additionally, we recognize the importance of clarifying key concepts, and we're considering the inclusion of a step-by-step tutorial for the IVRE game to facilitate better understanding.
>
> > It would be nice to include a discussion of the performance gap between search-naïve and the human performance.
>
> Thank you for your valuable suggestion. The comparison between these two baselines indeed presents an intriguing aspect of our study. There are multiple reasons for the Search-Naive agent's performance. One is that the agent instantiates ths assumption that the Blicket machine is an OR machine (if one is Blicket, the machine will be activated). The other is that while incorrect, the agent approximates causality with correlation, which could be partially right in some cases. The third is that the agent conducts Blicketness test one object at a time, which could help best disentangle other confonding factors, though inefficient. Compared to humans, Search-Naive does these few things good enough to reach fair performance. However, human subjects are much more efficient and have varied strategies in reaching conclusions, such as proposing multiple objects at a time to exclude several possibilities. We recognize the need for an expanded discussion in the revised version, and we are dedicated to providing deeper insights into this dynamic interplay between strategies and outcomes.
>
> > The environment is not terribly novel, as it largely adopts the framework of ACRE.
>
> While IVRE shares a setting akin to the Blicket experiment in ACRE, it's essential to emphasize the distinctive nature of IVRE: interactiveness. ACRE is primarily ***passive***, where the agent receives all information, whereas in IVRE, the agent needs to actively explore by itself to gather enough evidence.
> > Some more analysis regarding the very low visual environment performance would be helpful. While lower performance is to be expected, the numbers reported in the experiments section are surprisingly low.
>
> We also have noted the random-level performance of pixel-input models. We argue that the following reasons contribute to the low performance. For one thing, the hypothesis space in the visual space is much larger: machine activation could be related to object positionining, which is hard to be directly communicated to the learning agents using pixel input. For another, in pixel space, RL from few examples (meta learning) is widely known to be challenging and existing results reported in the surveyed papers (in Related Work) all show the same problem. We trained the agents using 10^7 steps and the learning curve showed convergence. There might be some problem structures that when incorporated into visual RL methods would show improvement compared to the current baseliens. However, for the first attempt, we begin with standard baselines to demonstrate the challenge.

---

> > ### Comment · Reviewer_KESZ · 2023-08-22
> >
> > Thank you for your response. A step-by-step tutorial would be a very helpful addition to the website. The reasons you provide for the surprising experimental results make sense, and I think that a sentence or two elaborating on these points in the paper would be beneficial as well. Aside from these amendments, I have no further comments or concerns.

---

> > > ### Author Response · Authors · 2023-08-24
> > >
> > > We sincerely appreciate your meticulous review and the constructive feedback you've provided. We have incorporated a detailed tutorial alongside the code and will release them in a github page. Further, we will include discussions in the rebuttal period into the manuscript to provide a more comprehensive understanding of the outcomes. Thank you once again for your time and valuable insights!

---

### Official Review · Reviewer_svXQ · 2023-07-23
**An interesting visual reasoning environment**

**Rating:** 7
**Confidence:** 4
**Correctness:** The claims seem correct.
**Clarity:** The paper is well-written.

**Strengths:**

- The environment seems like a good contribution. I could see people using it in future work.
- The paper is well-written.
- The paper is situated well with respect to prior work
- The baselines and experiments are comprehensive, and the analysis is useful

**Additional Feedback:**

N/A

**Documentation:**

There is sufficient documentation.

**Ethics:**

No ethics concerns.

**Limitations:**

Adequately discussed

**Opportunities For Improvement:**

I might have missed something, but it doesn't look like there is much discussion of the Search-Naive method's strong performance. Why is it far better than the other baselines?

Typos:
75: Serveing

**Relation To Prior Work:**

Prior work is adequately discussed.

**Summary And Contributions:**

This paper proposes IVRE, an RL environment for causal reasoning built on the CLEVR engine and designed to mimic Blicket experiments from cognitive psychology. The combination of elements in the proposed environment is novel and fills a gap in existing environments or datasets designed to measure visual causal reasoning. Numerous baselines are evaluated, and the results are discussed.

---

> ### Author Response · Authors · 2023-08-21
> **Rebuttal to reviewer svXQ**
>
> Dear reviewer svXQ,
>
> Thank you very much for your efforts in reviewing our paper!
>
> > I might have missed something, but it doesn't look like there is much discussion of the Search-Naive method's strong performance. Why is it far better than the other baselines?
>
> Thank you for your suggestion! We value your feedback, and in the revised version, we will certainly provide more in-depth explanations to enhance clarity. There are multiple reasons for the Search-Naive agent's performance. One is that the agent instantiates ths assumption that the Blicket machine is an OR machine (if one is Blicket, the machine will be activated). The other is that while incorrect, the agent approximates causality with correlation, which could be partially right in some cases. The third is that the agent conducts Blicketness test one object at a time, which could help best disentangle other confonding factors, though inefficient.
>
> > Typos
>
> Many thanks for pointing it out. We will fix this in the revised version.

---

### Official Review · Reviewer_hE8F · 2023-07-24
**IVRE**

**Rating:** 7
**Confidence:** 3
**Correctness:** Yes, as far as I can tell

**Strengths:**

The proposed benchmark is easy to understand and well described. The experiments are quite comprehensive. This benchmark is a useful and relevant demonstration that our current learning methods are insufficient for capturing the nuances of causal learning.

**Additional Feedback:**

Some questions for the authors:
* Is the randomized (physical) placement of objects in the visual environment a confounder that is making the learning task more difficult? Because one hypothesis of blicketness for these stimuli is the actual location of the objects. This doesn't appear to be a problem with the symbolic representation. Though confounders were mentioned briefly, maybe this one should be mentioned explicitly as well.
* Similarly, there was a brief discussion on hypotheses about properties of objects indicating (generalizable) blicketness. Is this ever studied as an actual hypothesis the system should learn? (Along those lines, is the physical location ever studied as a true hypothesis?)
* Is there an oracle for determining the best trials to propose? I suppose one of the rule-based / symbolic systems may include such an oracle, but I just don't know.
* What does it mean to "tame" LLMs (L245)? Maybe this is a typo

**Clarity:**

Yes, though a bit of re-ordering or formalization of defining the learning agent would be helpful

**Documentation:**

Yes

**Limitations:**

The only limitation in terms of social impact is the lack of details on the human study.

**Opportunities For Improvement:**

It would be nice to have a bit more discussion on how the ability to learn this kind of causal reasoning / perform this kind of hypothesis testing may be important for the systems we are building. While I don't doubt this is fundamentally necessary for our systems, some discussion of particular examples would be interesting (e.g., maybe linking to observed failures of LLMs in processing real language).

I was somewhat confused about the separation in beliefs and proposed trials in the agent setup. It became clear later on, but maybe formally defining the agent beforehand would be useful.

There need to be more details on the human study. E.g., how participants were recruited, what platform was used, etc.

**Relation To Prior Work:**

Yes

**Summary And Contributions:**

This paper proposes a new benchmark environment for studying causal learning, based on the work of Gopnik et al. in studying causal reasoning in children. The proposed benchmark includes both visual stimuli (rendered images) and symbolic representations of the stimuli used for learning. The paper includes numerous experiments of different rule-based, prompt-based, and other learning techniques (including reinforcement learning).

---

> ### Author Response · Authors · 2023-08-21
> **Rebuttal to Reviewer hE8F (part 1)**
>
> Dear reviewer hE8F,
>
> Thank you very much for your thoughtful review.
> > It would be nice to have a bit more discussion on how the ability to learn this kind of causal reasoning / perform this kind of hypothesis testing may be important for the systems we are building. While I don't doubt this is fundamentally necessary for our systems, some discussion of particular examples would be interesting (e.g., maybe linking to observed failures of LLMs in processing real language).
>
> Thank you for your suggestion! We appreciate the opportunity to provide our perspective on this topic. Our emphasis on endowing AI with human-like abilities to tackle uncertainty resonates strongly with the challenges confronting AI systems. Traditional AI models typically rely on training with ***static*** datasets, subsequently making predictions based on established patterns. And yet, an ideal intelligent agent shall be able to ***actively*** explore the environments, make use of newly collected evidence to inform its future actions, and potentially repeating if the earlier hypothesis is proven wrong.
> One perspective on LLMs is that LLMs behave as static soft databases. We can query from it, assuming that it has consumed and remembered sufficient related data. However, it lacks *agency* AGI requires and that has become a trending research direction.
> The capacity to actively resolve uncertainty assumes a pivotal role in granting AI systems agency. Imagine the following case: you returned home only to find your house in a mess. The ability to find out the reason is clearly beyong only language understanding. It could be possible that you just forgot to close the window and there was strong wind during the day. Or, some animals slipped in. Or in the worse case, a thief visited. To validate and confirm the hypothesis, an agent needs ***knowledge*** to formulate the possibilities, a simulator to simulate what a ***world*** would look like if each hypothesis happened, and correspondingly check the real world situation. These components involve complex cognitive tools working together, eg. memory, planning, simulating, etc. And language models nowadays are just incapable.
> In conclusion, the skill of actively addressing uncertainty contributes to the autonomy and effectiveness of AI systems, allowing them to navigate and comprehend complex, unfamiliar scenarios. Thank you again for initiating this discussion, and we welcome further engagement on this thought-provoking topic.
> > I was somewhat confused about the separation in beliefs and proposed trials in the agent setup. It became clear later on, but maybe formally defining the agent beforehand would be useful.
>
> Thank you for your suggestion. We will add formal definitions in the related sections.
> > There need to be more details on the human study. E.g., how participants were recruited, what platform was used, etc.
>
> Thanks for your suggestion! These details can be found in the supp Section E. The paper demonstrated pilot experiment results. We will add more detailed results to the final version.
>
> > Is the randomized (physical) placement of objects in the visual environment a confounder that is making the learning task more difficult? Because one hypothesis of blicketness for these stimuli is the actual location of the objects. This doesn't appear to be a problem with the symbolic representation. Though confounders were mentioned briefly, maybe this one should be mentioned explicitly as well.
>
> > Similarly, there was a brief discussion on hypotheses about properties of objects indicating (generalizable) blicketness. Is this ever studied as an actual hypothesis the system should learn? (Along those lines, is the physical location ever studied as a true hypothesis?)
>
> Thanks for your question. To clarify, we do not study position or other properties as indicators for blicketness. The assumption is clearly instantiated in our symbolic version (the input is one-hot vectors for different objects); however, it is not easy to do so in the pixel version. As a first step towards this problem, our aim was to maintain simplicity. Prior cognitive studies have explored confounding effects from object properties and positions [B and C], and we remain open to extending IVRE in these directions in the future.

---

> ### Author Response · Authors · 2023-08-21
> **Rebuttal to Reviewer hE8F (part 2)**
>
> > Is there an oracle for determining the best trials to propose? I suppose one of the rule-based / symbolic systems may include such an oracle, but I just don't know.
>
> From our point of view, we can hardly see a ***general*** oracle solution for the proposed IVRE. IVRE needs agents to propose efficient trials to resolve uncertainty in the current episode. The hypothesis space can be seen as a tree with $ 2^{objnum} $ branches. And aprior you know there are $ n $ blickets in total. In each step, we can prune some branches based on the new information and the known number of blickets. The most related math problem is Balance Puzzle. However, our problem differs from the puzzle in that there could be an arbitrary number of blicketks rather than 1, and that we do not need an equal number of objects to test in each step. By manual inspection, we also note that the full search process would be long and short for configurations with only minor differences, given different context. We believe the search process is ad-hoc. In human experiments, participants tend to employ an approximative version of extensive search, strongly relying on heuristics.
>
> > What does it mean to "tame" LLMs (L245)? Maybe this is a typo
>
> No, it's not a typo. In this context, "tame" is used to mean controlling, managing, or guiding in a specific manner (https://www.merriam-webster.com/dictionary/tame). We appreciate your observation that while this term has been utilized in prior algorithmic contexts, it could still potentially be unclear.
>
> [A] Tigas, Panagiotis, et al. "Interventions, where and how? experimental design for causal models at scale." Advances in Neural Information Processing Systems 35 (2022): 24130-24143.
> [B] Gopnik, Alison, and David M. Sobel. "Detecting blickets: How young children use information about novel causal powers in categorization and induction." Child development 71.5 (2000): 1205-1222.
> [C] Cook, Claire, Noah D. Goodman, and Laura E. Schulz. "Where science starts: Spontaneous experiments in preschoolers’ exploratory play." Cognition 120.3 (2011): 341-349.

---

### Author Response · Authors · 2023-08-21
**Overall rebuttal**

To all reviewers,

We sincerely appreciate and are grateful for the time reviewers have spent reading our work and providing useful, thoughtful and constructive feedback. We also gained many new insights to improve IVRE from the reviews. We wil further revise the paper based on the suggestion. We are very glad to see that there is a general interest and agreement on accpetance in our proposed IVRE benchmark. We hope that IVRE can serve as a valuable contribution to the whole community.

---

### Decision · Program_Chairs · 2023-09-22

**Decision:**

Accept (Poster)

**Comment:**

The authors have proposed an interactive environment titled IVRE featuring scenarios involving *Blicket* detection. The primary motivation is to provide a resource for evaluating the capability of agents to reason under uncertainty, which is a natural objective towards developing agents with humanlike intelligence. The reviewers have expressed that the paper is well-written, and the clarity and empirical rigor has further improved after the authors have incorporated the suggestions of the reviewers and the discussions during the rebuttal phase. The webpage is well documented and provides relevant details. I can see this being a resource that is useful for the community.

The code/doc and sample still point to GDrive links; I encourage the authors to heed the reviewers' suggestions and move to hosting the code on Github or equivalent, which would further help in the adoption of IVRE. The authors have expressed that they will do so, and I am taking this in good faith.

Given the favorable response of the reviewers, the updated and improved discussion and additional experiments from the authors, and the overall promising utility of this work to the community, I recommend Accepting this paper.